# Phenological Variations in the Content of Polyphenols and Triterpenoids in *Epilobium angustifolium* Herb Originating from Ukraine

**DOI:** 10.3390/plants13010120

**Published:** 2023-12-31

**Authors:** Liudas Ivanauskas, Kateryna Uminska, Zigmantas Gudžinskas, Michael Heinrich, Victoriya Georgiyants, Alla Kozurak, Olha Mykhailenko

**Affiliations:** 1Department of Analytical and Toxicological Chemistry, Lithuanian University of Health Sciences, A. Mickevičiaus Str. 9, 44307 Kaunas, Lithuania; 2Zhytomyr Basic Pharmaceutical Professional College, Chudnivska Str. 99, 10005 Zhytomyr, Ukraine; uminska.kateryna@pharm.zt.ua; 3Nature Research Centre, Institute of Botany, Žaliųjų Ežerų Str. 47, 12200 Vilnius, Lithuania; zigmantas.gudzinskas@gamtc.lt; 4Pharmacognosy and Phytotherapy Group, UCL School of Pharmacy, 29-39 Brunswick Square, London WC1N 1AX, UK; m.heinrich@ucl.ac.uk (M.H.); o.mykhailenko@ucl.ac.uk (O.M.); 5Chinese Medicine Research Center, College of Chinese Medicine, China Medical University, Taichung City 404, Taiwan; 6Department of Pharmaceutical Chemistry, National University of Pharmacy, 4-Valentynivska Str., 61168 Kharkiv, Ukraine; vgeor@nuph.edu.ua; 7Carpathian Biosphere Reserve, 90600 Rakhiv, Ukraine; akozurak@gmail.com

**Keywords:** biologically active compounds, *Chamaenerion angustifolium*, flowering phase, phenology

## Abstract

The composition of secondary metabolites undergoes significant changes in plants depending on the growth phase and the influence of environmental factors. Therefore, it is important to determine the harvesting time of plant material for the optimum secondary metabolite profile and therapeutic activity of the primary material. The shoots of *Epilobium angustifolium* are used as a healing tea due to the presence of polyphenolic compounds. The aim of this study was to assess the composition of phenolic compounds and triterpenoid saponins in *E. angustifolium* leaves and flowers and to estimate the dynamics of their content depending on the flowering phase. Qualitative and quantitative characterisation of polyphenols and triterpenoids in *E. angustifolium* samples from Ukraine of three flowering phases were performed using the high-performance liquid chromatography photo diode array (HPLC-PDA) method. During the present study, 13 polyphenolic compounds and seven triterpenoids were identified in the plant material. It was noted that the largest content and the best polyphenol profile was in late flowering. The most important polyphenolic compounds in the plant material were chlorogenic acid, hyperoside, isoquercitin, and oenothein B. The triterpenoid profile was at its maximum during mass flowering, with corosolic and ursolic acids being the dominant metabolites. The results of the analysis revealed that the quantity of many of the tested metabolites in the raw material of *E. angustifolium* is dependent on the plant organ and flowering phase. The largest content of most metabolites in the leaves was in late flowering. In the flowers, the quantity of the metabolites studied was more variable, but decreased during mass flowering and increased significantly again in late flowering. The results show that *E. angustifolium* raw material is a potential source of oenothein B and triterpenoids.

## 1. Introduction

*Epilobium angustifolium* L. [Onagraceae, *Chamerion angustifolium* (L.) Holub and *Chamaenerion angustifolium* (L.) Scop.], is a widespread and variable species commonly known as ‘fireweed’ in North America, ‘great willow herb’ in some parts of Canada, and ‘rosebay willow herb’ in Britain [1]. *Epilobium angustifolium* has a circumboreal distribution extending from arctic and subarctic regions south into cool-temperate regions of North America and Eurasia [2]. It grows mainly in forest and alpine grasslands, in semi-open mixed forests, and along forest edges and banks of rivers and streams. *Epilobium angustifolium* is especially common in recent forest clearings and burn sites [1].

*Epilobium angustifolium* is a traditional food and medicinal plant in Europe, especially in the northern and eastern countries [3]. The above-ground parts of *E. angustifolium* and other *Epilobium* species are eaten as a salad [3]. This species has been used as a poultice to treat mouth wounds in the traditional medicine of the Gwich’in people [4]. Due to its astringent and emollient properties, *E. angustifolium* infusion was included in the American herbal list in the 19th century as a remedy for gastrointestinal diseases [5].

The leaves and flowers of *Epilobium* L. species are a rich source of secondary metabolites, especially polyphenols, including flavonoids, phenolic acids, and tannins [5,6,7], which possess antioxidant, antimicrobial, anticancer, and anti-inflammatory properties [4]. In addition to polyphenols, there are lipophilic compounds such as steroid triterpenoids and fatty acids [7,8]. The macrocyclic ellagitannins, especially oenothein B, are considered to be the major active compounds determining the biological activity of *E. angustifolium* extracts [9]. Several triterpene acids were also found in the *E. angustifolium* raw materials, namely, oleanolic, pomolic, and ursolic acids [7]. The presence of both triterpenoids and polyphenols (especially oenothein B) in herbal raw material of *E. angustifolium* and their mutual potentiation determines the presence of pronounced antioxidant potential [10], which helps neutralize free radicals, protecting cells from oxidative damage and inflammation, which underlie all diseases [11].

Therapeutic properties of *E. angustifolium* extracts were reported in various in vitro and in vivo pharmacological studies [5,11,12,13,14]. The healing of *E. angustifolium* polyphenols works through several mechanisms, including direct killing of cancer cells and microbes, antioxidant effects, metal chelation, and anti-inflammatory immunomodulation [5,12,15,16]. The European Medicines Agency report (2016) [14] does not contain data regarding clinical trials of *E. angustifolium* extracts. However, in 2021, a monocentric, randomized, double-blind, placebo-controlled clinical trial was published [17] to evaluate the effect of fireweed extract (500 mg) in the treatment of benign prostatic hyperplasia in men. The studies were carried out on 128 patients for 6 months and positive results were obtained.

To effectively use herbal raw materials, it is necessary to optimize the plant harvesting time so that the accumulation of bioactive compounds is maximized. The composition and content of plant secondary metabolites are influenced by various factors: from the growth phase to post-harvest processing of raw materials, which should be controlled [18,19,20].

Among the most important tasks in preparing raw materials for medicinal plants is to ensure their high quality, which depends to a large extent not only on the technology used, but also on the provenance of the plant, habitat conditions, and the stage of development or phenological phase of the plant [21,22,23]. It is therefore particularly important to determine at which stage of growth the plants contain the highest amount of the expected biologically active metabolites [24], or when their qualitative composition is the most appropriate to achieve the best therapeutic results [25]. We formulated a working hypothesis that the highest content of biologically active compounds in *E. angustifolium* raw material is concentrated during the mass flowering period. With the aim of testing this hypothesis, we developed an objective to determine the qualitative and quantitative composition of polyphenols and triterpenoids in the leaves and flowers of *E. angustifolium* during the early flowering, mass flowering, and late flowering phases and to assess the changes in the content of these compounds in the plant parts during flowering.

## 2. Results and Discussion

### 2.1. Variation in the Total Content of Phenolic Compounds

The total content of phenolic compounds in *E. angustifolium* leaf and flower samples (measured with Folin–Ciocalteu reagent using a spectrophotometer in mg of gallic acid equivalent per g of dry weight; hereafter, mg GA/g) varied in quite a narrow range (Table 1). The total content of phenolic compounds in leaf samples ranged from 1.17 mg GA/g to 1.54 mg GA/g and increased towards late flowering. The total content in flowers ranged from 1.13 mg GA/g to 1.19 mg GA/g and was similar to the content in leaves. The total content of phenolic compounds correlated with the content of each individual metabolite, which was determined using the high-performance liquid chromatography (HPLC) method.

### 2.2. HPLC Metabolite Profiling

The results of the HPLC metabolite profiling analysis in *E. angustifolium* samples revealed thirteen polyphenols (Figure 1) and seven triterpenoids (Figure 2). In terms of phytochemical properties, 20 compounds were classified into different groups including flavonoids (*n* = 7), polyphenolic acids and their derivatives (*n* = 4), tannins (*n* = 2), and terpenoids (four triterpenoid acids and three neutral triterpenoids). The total amount of polyphenols and triterpenoids in the *E. angustifolium* extracts ranged from 2.45 mg/g dw to 86.92 µg/g dw and from 1.02 mg/g dw to 6.36 mg/g dw, respectively (Table 2). In general, phenolic compounds qualitatively and quantitatively were more abundant in the flowers, while triterpenoid compounds were more abundant in the leaves. Using high-performance liquid chromatography diode-array detection (HPLC-DAD) data analysis, the identified polyphenols and triterpene saponins were then quantified in *E. angustifolium* samples.

### 2.3. The Quantification of Core Metabolites

#### 2.3.1. Phenolic Acids

Qualitative changes in phenolic acids caused by phenological phases were practically absent in the tested samples of *E. angustifolium*; however, significant differences were observed in the quantitative content of the compounds. Gallic acid, ellagic acid, chlorogenic acid, and neochlorogenic acid were present in all samples, with chlorogenic acid always being the main phenolic acid during the entire flowering period. The maximum content of chlorogenic acid was observed in the leaves in the late flowering phase (2.60 mg/g). It is possible to observe the following dynamics of changes in the content of chlorogenic acid in flowers by phase: early flowering (0.91 mg/g), mass flowering (1.77 mg/g), and late flowering phase (1.55 mg/g). Overall, the quantity of chlorogenic acid was relatively high and stable throughout the growing season.

#### 2.3.2. Flavonoids

The change in co-occurrence and flavonoid profile was highest among the polyphenols studied. In this study, the presence of six flavonoids was found, among them rutin, hyperoside, isoquercitrin, quercitrin, afzelin, and guaijaverin. The total content of flavonoids identified with HPLC was significantly higher in samples of flowers (3.11 mg/g) and leaves (2.94 mg/g) collected during late flowering; samples of flowers collected during early flowering and the mass flowering phases contained slightly less of them, 1.86 mg/g and 1.90 mg/g, respectively.

The dominant flavonoid in the samples was isoquercitrin, but the total yield of phenolic compounds did not depend on its content and was determined by the total contribution of other components. The isoquercitrin content changed significantly in leaves during the changes in the flowering phases. In flowers, the level increased gradually. In leaves in the stage of late flowering, the increase in the level of isoquercitrin was found to be almost three times higher than the level during mass flowering.

#### 2.3.3. Tannins

Significant variability in the content of oenothein B and oenothein A was observed depending on the flowering phase (Table 2). The highest content of oenothein B (54.62 mg/g) was observed in leaves in late flowering and in flowers (50.26 mg/g) om early flowering. A slightly lower content of oenothein B was observed in flowers (45.95 mg/g) in late flowering and, in general, the raw material harvested in this phase had the highest total tannin content (31.81 mg/g). The result was statistically significant and clearly indicated differences between the remaining samples at this stage. Rhizomes had the lowest accumulation of oenothein A and oenothein B, 0.13 and 1.95 mg/g, respectively. Among all tested samples, the tannin content was always higher in flowers (ranging from 39.70 mg/g to 45.95 mg/g) than in leaves (ranging from 15.45 mg/g to 24.90 mg/g), regardless of the flowering phase. The exception was leaves during late flowering, when the content of oenothein B was 54.62 mg/g.

#### 2.3.4. Triterpenoids

The triterpene acid profile consisted of maslinic, corosolic, oleanolic, and ursolic acids. The content of each of these acids was found to be higher in the leaves than in the flowers of *E. angustifolium* (Table 2). Ursolic acid and corosolic acid were the main triterpene acids in all samples, regardless of the phase of raw material harvesting. Ursolic acid dominated among all triterpenoids and its content was particularly high (ranging from 3.12 mg/g to 3.51 mg/g) in leaf samples regardless of the flowering phase. Although oleanolic acid is of great pharmacological importance, its content in *E. angustifolium* samples was slightly lower than corosolic and ursolic acids and ranged from 0.10 mg/g to 0.61 mg/g. Oleanolic acid was the only metabolite that dominated in the *E. angustifolium* rhizomes (0.51 mg/g). The accumulation of corosolic acid increased from the early flowering stage to the mass flowering phase of *E. angustifolium* and the content of the metabolite decreased sharply (from 1.32 mg/g to 0.12 mg/g). The leaf sample in early flowering had the richest triterpenoid profile, with all metabolites identified (maslinic acid, corosolic acid, oleanolic acid, ursolic acid, betulin, erythrodiol, and uvaol).

### 2.4. Validation of the HPLC Method

The developed HPLC-DAD method for the identification of polyphenols in *E. angustifolium* samples was fully validated. The calibration curve, limits of detection (LOD), limits of quantification (LOQ), and linear range for each analyte are provided in Table 3. All metabolites showed good linearity (r^2^ ≥ 0.997) within the tested ranges. The repeatability was expressed as the relative standard deviation (%RSD) of the major constituents’ content and the RSD ranged from 0.1% to 0.8%, which was satisfactory. The determination of the main metabolites in the tested solutions was carried out by comparing the peaks’ retention times and the UV spectrum obtained from the chromatogram of the standard solution. All results revealed the repeatability, accuracy, high sensitivity, and good linearity of the method.

### 2.5. Chemical Antioxidant Profile

Considering that plant extracts contain many different classes and types of antioxidants, we used different methods to evaluate the antioxidant profile. Methanol extracts of *E. angustifolium* leaves in the early flowering phase and flowers in the mass flowering phase showed the highest total content of phenols (1.29 mgGA/g and 1.19 mgGA/g dw, respectively), which correlates with the content of individual phenolic compounds (Table 1). For the determination of antiradical effects, the ABTS radical scavenging activity method and ferric reducing antioxidant power (FRAP) assay in vitro were used. The ABTS•+ free radical scavenging effect of *E. angustifolium* extract varied considerably: from 9755.56 μg TE/g to 2577.78 μg TE/g. The highest ABTS radical–cation binding effect was observed in the leaves collected in the early flowering phase at 9755.56 μg TE/g. The antioxidant profile found with FRAP (11,677.51 mg FE(II)/g dw) for leaves during the early flowering phase (9097.50 mg FE(II)/g dw) and for leaves during the mass flowering phase was also the highest, and the lowest reducing profile was found in the flowers in the early flowering phase (2287.5 mg FE(II)/g dw). In general, the antioxidant potential of *E. angustifolium* raw material throughout the entire flowering period was high, while the leaves had a higher efficiency than the flowers. Both methods showed that leaves and flowers had slightly higher activity during the late flowering phase.

### 2.6. Effects of the Flowering Phase on the Content of Metabolites

#### 2.6.1. Quantitative Dynamics of Polyphenols

The analysis of polyphenol dynamics in relation to the flowering phase, plant organ, and the interaction of these two factors showed that all the studied metabolites, except quercetin, were significantly affected by both factors and their interaction (Table A1). The plant part did not have a significant effect (*p* = 0.913) on the quercetin content, but the interaction between the flowering phase and the plant part had a significant effect (*p* < 0.001) on its content.

The analysis of the polyphenolic compound content in *E. angustifolium* leaves and flowers at different flowering phases showed that the dynamics of nine metabolites (afzelin, chlorogenic acid, ellagic acid, gallic acid, isoquercitrin, neochlorogenic acid, oenothein A, oenothein B, and quercitrin) had a similar pattern of variation. The content of all these metabolites was higher in flowers than in leaves. Furthermore, the content of all nine metabolites increased from early flowering to mass flowering in the flowers and decreased in the leaves. Thereafter, the dynamics of the metabolites in the plant parts was reversed (Figure 3). It was found that from mass flowering to late flowering, the content of these metabolites decreased significantly in the flowers and increased significantly in the leaves. The content of two polyphenols, hyperoside and rutin, was lowest during early flowering and then increased steadily and significantly (*p* < 0.05). Moreover, the content of these metabolites did not differ significantly (*p* > 0.05) between flowers and leaves during the same flowering phase (Figure 3). The dynamics of quercetin in *E. angustifolium* leaves and flowers differed from the dynamics of the other studied phenolic compounds. During early flowering, the content of quercetin in leaves and flowers was similar (*p* = 0.913) but decreased in leaves during mass flowering and increased significantly (*p* < 0.001) by late flowering. In flowers, the quercetin content increased consistently with changing flowering phases. The pattern of guaijaverin dynamics was quite peculiar. During early flowering, the guaijaverin content in flowers and leaves was low, but by mass flowering it increased significantly (*p* < 0.001). Later, until late flowering, the guaijaverin content in leaves increased and in flowers decreased significantly (*p* < 0.01).

The total content of polyphenolic compounds in *E. angustifolium* flowers was significantly higher (*p* < 0.001) than in leaves during the early flowering and mass flowering phases, while in late flowering their content was significantly higher in leaves (*p* < 0.001) than in flowers (Figure 3).

#### 2.6.2. Quantitative Dynamics of Triterpenoids

The analysis of the dependence of the triterpenoid content changes in leaves and flowers of *E. angustifolium* on the flowering phase, the plant organ, and the interaction between these two factors showed that the content of most metabolites is significantly affected by both factors and their interaction (Table A2). Nevertheless, some factors did not have a significant effect on the content of individual triterpenoids. It was found that the content of maslinic acid was not affected by the plant part, but the interaction between the flowering phase and the plant part had a significant effect on its content. None of the factors analysed (*p* = 0.134 for the effect of the plant part and *p* = 0.266 for the effect of the flowering phase) or their interaction (*p* = 0.266) had a significant effect on the erythrodiol content (Table A2). In addition, the total triterpenoid content was significantly affected by the plant part and flowering phase, but the interaction between the two factors was not significant (*p* = 0.164).

The analysis of the triterpenoid content in *E. angustifolium* parts in different flowering phases revealed that the dynamics of the individual metabolites exhibit specific trends. The content of maslinic acid was highest during early flowering, but decreased significantly in both leaves (*p* < 0.001) and flowers (*p* < 0.001) by mass flowering and further declined insignificantly (*p* = 0.756) by late flowering. A similar trend was also observed for oleanolic acid content changes in leaves and flowers of *E. angustifolium* (Figure 4). A different pattern of corosolic acid content changes in *E. angustifolium* leaves and flowers was observed. Its content in leaves and flowers increased significantly from early flowering to mass flowering (*p* < 0.001), while the content decreased significantly (*p* < 0.001) until late flowering (Figure 4). The pattern of ursolic acid and betulin content in the plant parts was also different. The ursolic acid content in leaves varied insignificantly (*p* > 0.05) between flowering phases, while in flowers it increased significantly (*p* < 0.001) from early flowering to mass flowering and decreased significantly (*p* < 0.001) up to late flowering. During late flowering, the ursolic acid content was like (*p* = 0.915) that during early flowering (Figure 4). The content of betulin in the leaves was highest during early flowering and decreased significantly (*p* < 0.001) until mass flowering, although it remained almost unchanged (*p* > 0.05) in the flowers.

The total triterpenoid content in the leaves of *E. angustifolium* was significantly higher (*p* < 0.001) than in the flowers in all flowering phases. Furthermore, the content of triterpenoids in leaves and flowers was significantly higher during mass flowering than during early flowering (*p* < 0.001) and late flowering (*p* < 0.001).

## 3. Discussion

Plant phenolic compounds (e.g., flavonoids, phenolic acids) possess antioxidant properties, which means that they can neutralize harmful free radicals in the body that can cause oxidative damage to cells [26] and prevent to the development of chronic diseases, such as cancer, diabetes [27], and cardiovascular disease [28].

Thus, it is important to evaluate the antioxidant effects and phenolic content of each sample of *E. angustifolium* depending on the growing season to optimize the stage of collection of raw materials with the maximum content of metabolites. The total content of phenolic compounds in samples, expressed as gallic acid equivalents, ranged from 0.5585 mgGA/g dw (in rhizomes) to 1.2891 mgGA/g dw (in leaves in the early flowering phase). The total content of phenols in *E. angustifolium* samples remained relatively stable throughout the flowering season (Table 1). These data are consistent with the work of Jürgenson and Raal [29], who analysed the amount of phenolic compounds in *E. angustifolium* from Estonia. Therefore, leaves and flowers harvested from the end of June to the beginning of July are the best choice for *E. angustifolium* plant material with a consistently high content of polyphenols. The plant extract possessed antioxidant effects, probably due to the phenolic compounds present (oenothein B, chlorogenic acid, hyperoside, isoquercitrin). This could be explained by the fact that phenolic compounds are potent antioxidants due to their high redox potential allowing them to become hydrogen donors and singlet oxygen quenchers.

Using the HPLC method, the quantity of phenolic compounds and triterpenoid saponins in the raw material of *E. angustifolium* was determined. The quantitative content of phenolic compounds and triterpenoids in the samples studied is presented in Table 2. According to the studies of Jinting Li and Zhenghai Hu [30], the main synthesis of triterpene saponins in plants occurs in the palisade parenchyma, and the movement or transport of these metabolites occurs along the stem. In the samples tested, triterpenoids also accumulated more in *E. angustifolium* leaf tissue than in flowers, regardless of the flowering phase. Rutin, hyperoside, isoquercitrin, quercetin, oenothein A, oenothein B, and all polyphenolic acids were among the most abundant metabolites. Gallic acid, ellagic acid, quercetin, and kaempferol were frequently determined in the leaves and upper part of *E. angustifolium* shoots [5,7,15,31]. These flavonoids with antioxidant and anti-inflammatory properties are responsible for the pharmacological potential of the plant [4,15,16]. It should be noted that the poorest profile of metabolites was in *E. angustifolium* rhizomes. Since this type of raw material has a rather complex harvesting method, which is harmful to the plant, and at the same time the content of metabolites is exceptionally low, this provides justification for not harvesting rhizomes in the future.

According to the research of Mariola Dreger and coauthors [32], the highest level of chlorogenic acid was observed in the first two phases, i.e., in the vegetative and early flowering stages, whereas they observed a gradual decrease from the full flowering phase to the green fruiting phase. In Ukrainian samples, the content of chlorogenic acid also increased by stage and was maximum during late flowering in the leaves. High-performance liquid chromatography coupled to electrospray ionisation and quadrupole time-of-flight mass spectrometry (HPLC-ESI-QTOF-MS/MS) analysis of *E. angustifolium* from Poland [33] showed that the content of the metabolites was dominated by afzelin 148.2 μg/mL and neochlorogenic acid 37.3 μg/mL, as well as chlorogenic acid 20.8 μg/mL and isoquercetin 15.6 μg/mL. Unfortunately, we cannot convert the authors’ data into mg/g to compare with our results, since they did not indicate the initial weight of the extract used in the experiments (only a 1:10 plant-to-solvent ratio (m/v). Even though Poland and Western Ukraine are bordering regions, the climate of Ukraine is drier, which probably causes some differences in the qualitative composition of the metabolites.

The most common acids identified in *E. angustifolium* were gallic acid, ellagic acid, protocatechuic acid, vanillic acid, caffeic acid, and ferulic acid [32,34]. Analysing their presence and concentration across different growth stages of *E. angustifolium* could provide insights into variations in the secondary metabolites in plants during their development.

The flavonoid content in the samples was particularly high, especially in the leaves during late flowering. We found a positive relationship between rutin levels and isoquercitrin levels in *E. angustifolium* samples. These results agree with the findings of Valentová et al. [35] that isoquercitrin (a derivate of 3-O-glucosyltransferase) is an intermediate product of rutin biosynthesis in plant tissues. Isoquercitin is formed from rutin under the action of L-rhamnosidase, which in turn is converted into quercetin under the influence of β-D-glucosidase [36]. Hyperoside and quercetin can also be converted into each other [37], as can be observed by the quantitative content of these metabolites in the tested samples. In each sample, the content of hyperoside greatly prevailed over the content of quercetin. Leaves (2.00 mg/g) and flowers (1.38 mg/g) in the late flowering phase had a higher content of hyperoside in comparison with other phases, which again emphasises the optimality of this phase for raw material procurement in the Ukrainian Carpathians.

Oenothein is a natural tannin group with the main representatives oenothein A, oenothein B, and dimeric macrocyclic ellagitannin [38] and the main quality marker compound in Epilobium species. The presence of twenty-two phenolic hydroxyl groups in the structure of oenothein contributes to the strong antioxidant effects of plants.

The shikimate pathway for the biosynthesis of phenolic compounds, as well as the pathways for the biosynthesis of hydroxybenzoic acids (gallic acid and ellagic acid), has been described by many researchers [39,40]. There are several articles describing the biosynthesis of ellagitannins (tellimagrandin II, casuarictin, pedunculagin, 1-O-galloylglucose, punicalagin, urolithin A-C, isourolithin A) in leaves of Quercus robur L. [41], Punica granatum L. [42], Camellia sinensis (L.) Kuntze [43], etc. But at the same time, there is no clear understanding of the biosynthesis of oenothein B in *E. angustifolium*. Therefore, considering the generally accepted trends in the biosynthesis of phenolic compounds, we proposed a possible route of biosynthesis of oenothein B in Epilobium raw materials (Figure 5).

The content of oenothein B increased gradually and reached a maximum in the late flowering phase of *E. angustifolium* samples from Ukraine. An increase in the oenothein B content in *E. angustifolium* leaves during the growth phase was also observed by other authors [31,32]. Since gallic and ellagic acids, as derivatives, probably play a major role in the biosynthesis pathway of oenothein A and oenothein B, the relationship was also traced: in samples that had a higher content of gallic acid (0.64 mg/g in *E. angustifolium* leaves samples in the late flowering phase), the content of oenothein B was highest (54.62 mg/g).

Pentacyclic triterpenoids, oleanolic and ursolic acids particularly, exhibit pronounced anti-inflammatory and antioxidant effects [44] and probably have a certain contribution to this activity in Epilobium raw materials [5].

There is no direct connection between phenolic compounds (shikimate pathway) and triterpenoid saponins (isoprenoid pathway); they have two different biosynthesis pathways. However, Geană and coauthors [45] have found certain relationships. They have revealed that apple varieties with a higher content of phenolic compounds contain less triterpene saponins. In *E. angustifolium* samples, the total content of triterpenoids ranged from 1.02 mg/g to 6.36 mg/g, which was significantly less than the total content of all phenolic compounds in the samples (which ranged from 2.45 mg/g to 86.92 mg/g).

Correlation analysis showed the dynamics of the distribution of various polyphenolic compounds in *E. angustifolium* at different stages of the flowering phase and in different plant organs. In general, the content of the studied polyphenolic compounds, except for quercetin, was higher in flowers compared to leaves. The content of the metabolites included increased from early flowering to mass flowering for flowers, but decreased in leaves. This trend is likely due to several factors, including physiological changes in the plant during the flowering phase and its metabolism [46,47]. From mass flowering to late flowering, the opposite situation occurred: the quantity of metabolites in Epilobium samples decreased in flowers and increased in leaves. Hyperoside and rutin showed a unique pattern, gradually increasing from early to late flowering in both flowers and leaves. Thus, it was established that initially, during the period of early and mass flowering, flowers contained significantly more total polyphenolic compounds compared to leaves. However, in the late flowering phase, the content of these metabolites in the leaves became higher than in the flowers.

The dynamics of triterpenoids was different. For example, not only is the maslinic acid content influenced by the plant part, but the interaction between flowering phase and plant part has a significant influence. This suggests that the synthesis or accumulation of maslinic acid may be regulated by the combined action of these factors. The total triterpenoid content was influenced by both plant part and flowering phase, indicating different concentrations of these metabolites depending on the developmental stage of the plant and the organ in question.

The maximum content of maslinic and oleanolic acid was observed during early flowering, decreasing towards mass and late flowering in both leaves and flowers. Studies of triterpenoids in other plants have shown similar phase-dependent changes in content at different developmental stages. For example, studies on Betula species [48] revealed differences in betulin content at different growth stages, which is consistent with our results in *E. angustifolium*. This study revealed that betulin content had different trends in leaves and flowers, with maximum levels in the early flowering phase in leaves and an almost-constant level in flowers. The consistently higher content of triterpenoids in leaves than in flowers throughout the flowering phases may indicate a specific metabolic role or physiological significance of triterpenoids in leaves during the growth cycle of *E. angustifolium*. Thus, these results highlight the complexity of triterpenoid dynamics in different parts of *E. angustifolium* during different flowering phases.

During certain phases (especially late flowering), the total polyphenol content and triterpenoids varied significantly between leaves and flowers, indicating a dynamic shift in the accumulation of phenols during the growing season of a plant. In the future, it is necessary to conduct an additional analysis of phenolic compounds’ relationships with environmental factors in order to find out which external factors have a greater effect on the dynamics of polyphenols during the growing season of the plant [49]. However, our study only covers a relatively short period (25 days) and a longer period of analysis would be needed to shed more light on this, especially during the rapid growth period during spring.

In general, the assessment of the dynamics of phenolic compounds depending on the plant organ and phase can provide a deeper understanding of the interaction of bioactive metabolites in the leaves and flowers of *E. angustifolium* at different stages of the flowering season, as well as optimise the timing of raw material collection.

## 4. Materials and Methods

### 4.1. Plant Collection and Authentication

The samples of *E. angustifolium* L. (*Chamaenerion angustifolium* (L.) Scop [50]; Onagraceae) were collected in Western Ukraine, Transcarpathian region, Carpathian Mountains, Chornohirsky massif (altitude 1800 m above sea level; 48.047151° N, 24.631051° E) at the foot of the town of Pip Ivan Chornohiskyi in 2019. The species was identified and authenticated by Dr. Kozurak (Carpathian Biosphere Reserve, Rakhiv, Transcarpathian Region, Ukraine) and deposited in the Scientific Herbarium of the Carpathian Biosphere Reserve (CBR), Ukraine (voucher specimens CBR3328, CBR3556, CBR3798). Geographic coordinates were determined using a GPS device (Prestigio GeoVision 5056). The meteorological conditions of the material collection site were described following the data of the Ukrainian Hydrometeorological Centre.

The Ukrainian Carpathians lie in the European Continental climatic zone, the main features of which are determined by the predominance of the Atlantic Ocean air masses and transformed continental air masses. The Carpathians are characterised by a temperate continental climate, with high humidity, erratic springs, mild summers, warm autumns, and mild winters [51]. The Ukrainian Carpathians, according to the thermal altitudinal zonation, are classified as moderately cold (1250–1500 m) and cold (1500–2000 m), with mean July temperatures ranging from +12 °C to +8–9 °C and January temperatures ranging from −10 °C to −12 °C. The climatic conditions in the mountains are optimal for the growth of *E. angustifolium*. The cooler climatic conditions resulting from the altitude lead to a later flowering onset of *E. angustifolium* and a longer duration of the individual phenological phases compared to lowland plants. Although *E. angustifolium* grows in different regions of Ukraine [52,53], the largest populations are found in the Carpathians.

The samples of *E. angustifolium* used in this study were collected from a wild population located in a recent forest clearing on a gentle slope. The raw material of *E. angustifolium* was collected considering the plant flowering phase. Three flowering phases were distinguished: early flowering, mass flowering, and late flowering. Plants in the early flowering phase were sampled on 23 June, when only single flowers (less than 5%) in the lower part of the inflorescence were open, and the remaining flowers were in the bud stage. Plants in the mass flowering phase were harvested on 4 July, when more than half (at least 50%) of the total number of flowers in the inflorescence were open, with only the uppermost flowers still in the bud stage. Plants in the late flowering phase were harvested on 18 July, when the uppermost flowers of the inflorescence were open, and fruits (from at least 5% of lowermost flowers) had already developed at the base of the inflorescence. The upper 1/3 part of the plants in each flowering stage was sampled. One sample consisted of 25 plants collected randomly over the whole area of the stand. Once the samples were collected, the inflorescence and leaves were immediately separated. The inflorescence and the leaves were separated at the level of the lower flower.

### 4.2. Chemicals and Solvents

Analytical- and chromatographic-grade chemicals and solvents were used for this study: acetonitrile, methanol, glacial acetic acid, erythrodiol, maslinic, oleanolic acids from Sigma-Aldrich (GmbH, Karlsruhe, Germany); uvaol, betulin, corosolic acids, ursolic acid from Carl Roth (Karlsruhe, Germany); polyphenols (gallic acid, ellagic acid, neochlorogenic acid, chlorogenic acid, oenothein B, rutin, hyperoside, isoquercitrin, quercitrin, quercetin, guaijaverin, afzelin) were purchased from ChromaDex (Santa Ana, CA, USA), Sigma-Aldrich (Saint Louis, MO, USA), and Hwi Analytik (Rülzheim, Germany); all solvents used were of HPLC grade. Water was obtained using a Mili-Q purification system (Millipore, Burlington, MA, USA). Also, other reagents were used, such as sodium carbonate (Sigma-Aldrich, Scnelldorf, Germany), 2,2′-azino-*bis*(3-ethylbenzothiazoline-6-sulfonic acid) (ABTS), 2,4,6-tris(2-pyridyl)-s-triazine (TPTZ), aluminium chloride, hexaethylenetetraamine, acetic acid obtained from Sigma-Aldrich (Buchs, Switzerland); potassium persulfate from Alfa Aesar (Karlsruhe, Germany); Trolox (98%) was received from Fluka Chemika (Buchs, Switzerland).

### 4.3. Preparation of Extracts

The collected raw material was dried at an ambient temperature of 20–24 °C and used for the chemical analysis. Dried samples (15 g of each sample) were crushed to obtain particles of 2–3 mm size. For analysis of polyphenols, a precise weight (0.2 g) of each sample powder was extracted in 10 mL of 50% (*v*/*v*) methanol in an ultrasonic bath (WiseClean) at 45 ± 2 °C for 20 min. For triterpenoid analysis, 1.0 g of each powdered sample was filled with 1 mL 100% (*v*/*v*) acetone and subjected to ultrasound-assisted extraction for 30 min at room temperature. The extractive solutions were then centrifuged for 30 min at 3000× *g* in a Biofuge Stratos centrifuge. The obtained extracts were filtered through 0.45 µm syringe filters (Carl RothGmbH & Co. KG, Karlsruhe, Germany) and stored at 4 °C until the analysis.

### 4.4. Standard Solutions

Stock solutions of all triterpenoids were prepared in methanol at 200 µg/mL and stored at −20 °C until use. A serial dilution was made on each stock solution with methanol to prepare working standard solutions at concentrations in the linear range. Combined working solutions of standards of triterpenoids were obtained by mixing the mixing stock solutions.

### 4.5. Determination of Total Phenol Content

For the determination of the total polyphenol amount, the 50% (*v*/*v*) methanolic extract was mixed with the Folin–Ciocalteu phenol reagent, 9 mL of 7% sodium carbonate (Na_2_CO_3_) was added, and the mixture was kept in a dark place for 90 min. The absorbance of solutions was measured at 750 nm using a Halo DB-20 UV-Vis spectrophotometer (Techcomp Europe, UK). The obtained data were evaluated according to the linear regression equation of the end acid calibration graph: y = 0.9068x + 0.0617; *R*^2^ = 0.9960; y = absorption intensity; x = total phenolic compounds expressed as gallic acid equivalent per gram dry weight (mg GAE/g dw) [10].

### 4.6. HPLC Chromatographic Analysis

Chromatographic separation of polyphenols and triterpenoids was performed using the Waters e2695 Alliance HPLC system coupled with a 2998 PDA detector (Waters, Milford, MA, USA). Phenolic compounds were separated on an ACE Super C_18_ (250 mm × 4.6 mm, 3 µm) column (ACT, Aberdeen, UK) with a full run time of 81 min. Column temperature was 25 °C. The gradient elution mode consisting of 0.1% (*v*/*v*) trifluoroacetic acid in pure water (A) and acetonitrile (B) was as follows: 0 min, 5% B; 8–30 min, 20% B; 30–48 min, 40% B; 48–58 min, 50% B; 58–65 min, 50% B; 65–66 min, 95% B; 66–70 min, 95% B; 70–81 min, 5% B. The flow rate was 1.000 mL/min, and the injection volume was 10 µL. Peaks were found by comparing the UV-Vis spectra of each peak to valid reference standards and measuring their retention times. The samples were subjected to two different analyses. The quantity of oenothein A was obtained by recalculation through oenothein B.

For chromatographic analysis of triterpenoids, the same chromatography system was used, but the mobile phase consisted of acetonitrile and water (89:11, *v*/*v*) delivered at a flow rate of 0.7 mL/min in the isocratic mode. The column temperature was set at 15 °C with an injection volume of 10 µL. Detection of all triterpenoids was performed at a wavelength of 205 nm, corresponding to the maximum absorption, and peaks were identified with retention times as compared with standards. Quantification of triterpenoids was established by applying the external standard method, according to previous studies [10,54].

### 4.7. Validation of the Analytical Method

The analytical HPLC-PDA method for polyphenols was validated according to the ICH Q2 (R1) guidelines. Identification was performed by scanning in a range of 200–400 nm wavelengths by comparing spectral data and retention times to those of standard compounds. For quantification, 5–7-point linear calibration curves (r > 0.999) were constructed by plotting the response of each analyte, considered by target concentrations (in the range of 1.6–200.0 µg/mL). Limits of detection (LOD) and quantification (LOQ) were determined using the signal-to-noise ratio method. Precision values (repeatability of replicates on the same day and intermediate precision on three consecutive days), expressed as percentage relative standard deviations (% RSD) of peak areas, did not exceed the 2% threshold. Percent recoveries of studied phenolic compounds were in the acceptable range of 90–110% for our studied concentration levels, showing the trueness of the method. The quantity of metabolite was calculated from an external standard calibration in the concentration range of 0.5–100.0 µg/mL (*r*^2^ = 0.997). Each sample was analysed twice, and the mean value was used for calculation.

### 4.8. Antioxidant Profile

#### 4.8.1. ABTS Radical Scavenging Assay

The spectrophotometry analysis with ABTS was conducted as follows: 10 µL of the 50% (*v*/*v*) methanolic plant extract was mixed with 3 mL of working ABTS•+ stock solution (concentration was 2 mmol/L). The mixture was kept in the dark for 30 min, then its absorption was measured with a Halo DB-20 UV-Vis spectrophotometer (Techcomp Europe, UK) at a wavelength of 734 nm [26,55]. The calibration graph was created using standard Trolox solutions of 8000 to 24,000 µmol/L: y = 9 × 10^−5^x − 0.0081; *r*² = 0.9989; y = extent of absorption; x = antioxidant. The results are reported regarding the Trolox equivalent per gram of dry weight (TE/g dw).

#### 4.8.2. FRAP Radical Scavenging Effects Assay

The FRAP test was performed by combining 0.3 M acetate buffer, 10 mM TPTZ solution, 20 mM HCl, and 20 mM ferric chloride solution. After that, 10 µL of the sample was combined with 3 mL of the FRAP reagent, and the resulting mixture was well combined. After 30 min incubation, the absorption was measured at 593 nm using a Halo DB-20 UV-Vis spectrophotometer (Techcomp Europe, UK). Ferrous sulphate was used to obtain the calibration curve, which had the following equation: y = 0.0002 x + 0.0135; *R*^2^ = 0.9915 [26]. The results are expressed as ferrous sulphate equivalent per gram dry weight ((FE(II)/g dw). Each measurement was performed three times.

### 4.9. Statistical Analyses

The data were processed using the Microsoft Office Excel 2010 (Microsoft, JAV) software package. All data processing was carried out using the LabSolutions Analysis Data System (Shimadzu Corporation). The results of descriptive statistics are presented as mean and standard deviation (mean ± SD) from three replicates (*n* = 3) for each sample. The value of *p* < 0.05 was taken as the significance level. Linear regression was used to calculate the determination coefficients (*r*^2^) of the regression lines for each quantified metabolite.

The effect of plant organ and phenological phase on the content of the analysed metabolites in the *E. angustifolium* raw material was assessed with two-way fixed effect ANOVAs. Plant organ (leaves and flowers) was considered a fixed factor and the phenological phase (early flowering, mass flowering, and late flowering) was considered a random factor. To assess differences between pairs of multiple samples, the non-parametric Kruskal–Wallis test was applied. The calculations were computed and the graphs were plotted using the PAST 4.13 software [56].

## 5. Conclusions

This study presents the results of a profile of polyphenolic compounds and triterpenoids in the aerial parts of *E. angustifolium* in three flowering phases in Ukraine. Chromatographic analysis (HPLC-FDA) showed the presence of five triterpenoid acids and eight neutral triterpenoids in plant material. The results showed the promise of using *E. angustifolium* raw material not only as a source for oenothein B with anticancer activity, but also as a promising source of triterpenoids. The content of polyphenols and antioxidant effects in *E. angustifolium* extracts showed significant fluctuations depending on the flowering phase, and the nature of the changes was similar. The content of phenolic acids, flavonoids, and tannins in the extracts of the plant aerial parts was closely related to antioxidant activity, indicating that extracts with a higher content of polyphenols have higher antioxidant effects. Moreover, the antioxidant capacity of phenols depends on the location of the functional group in the core structure. The harvest time significantly affected the content of phenols in the leaves and flowers of *E. angustifolium* and, consequently, the antioxidant effects. Based on the results, we suggest that the late flowering phase is the best time to harvest herbal raw material. As a potential source of natural antioxidants, *E. angustifolium* could be a particularly useful addition to pharmaceuticals and functional food ingredients in both the nutraceutical and food industries.

## Figures and Tables

**Figure 1 plants-13-00120-f001:**
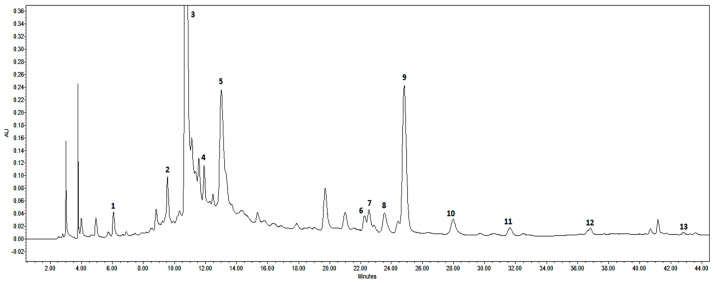
Representative HPLC-DAD chromatograms of polyphenols of *Epilobium angustifolium* flowers harvested during early flowering. Peak assignments: 1—gallic acid; 2—neochlorogenic acid; 3—oenothein B; 4—chlorogenic acid; 5—oenothein A; 6—ellagic acid; 7—rutin; 8—hyperoside; 9—isoquercitrin; 10—guaijaverin (quercetin-3-O-arabinopyranoside); 11—quercitrin; 12—afzelin (kaempferol-3-O-rhamnoside); 13—quercetin.

**Figure 2 plants-13-00120-f002:**
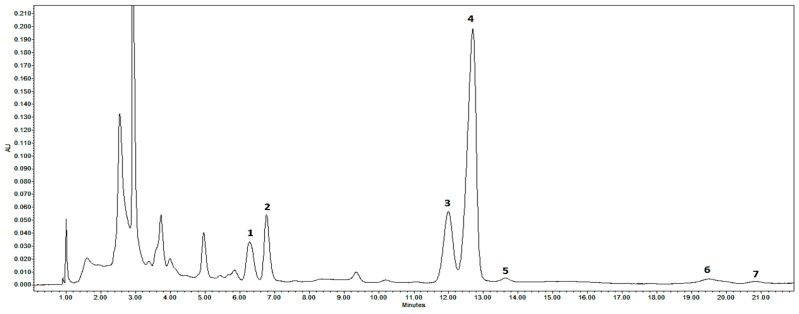
Representative HPLC-DAD chromatograms (λ = 205 nm) of triterpenoid acids and neutral triterpenoids of *Epilobium angustifolium* flowers harvested during early flowering. Peak assignments: 1—maslinic acid; 2—corosolic acid; 3—oleanolic acid; 4—ursolic acid; 5—betulin; 6—erythrodiol; 7—uvaol.

**Figure 3 plants-13-00120-f003:**
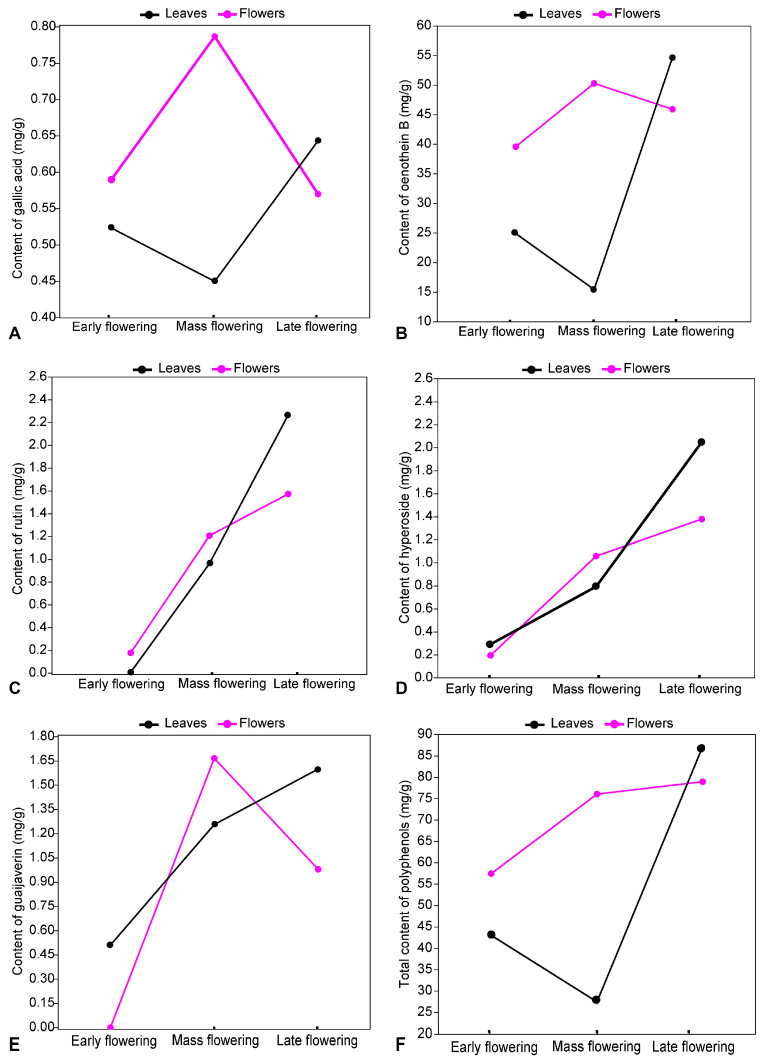
Dynamics of gallic acid (**A**), oenothein B (**B**), rutin (**C**), hyperoside (**D**), and guaijaverin (**E**) and the total content of polyphenols (**F**) in *Epilobium angustifolium* flowers and leaves in different flowering phases.

**Figure 4 plants-13-00120-f004:**
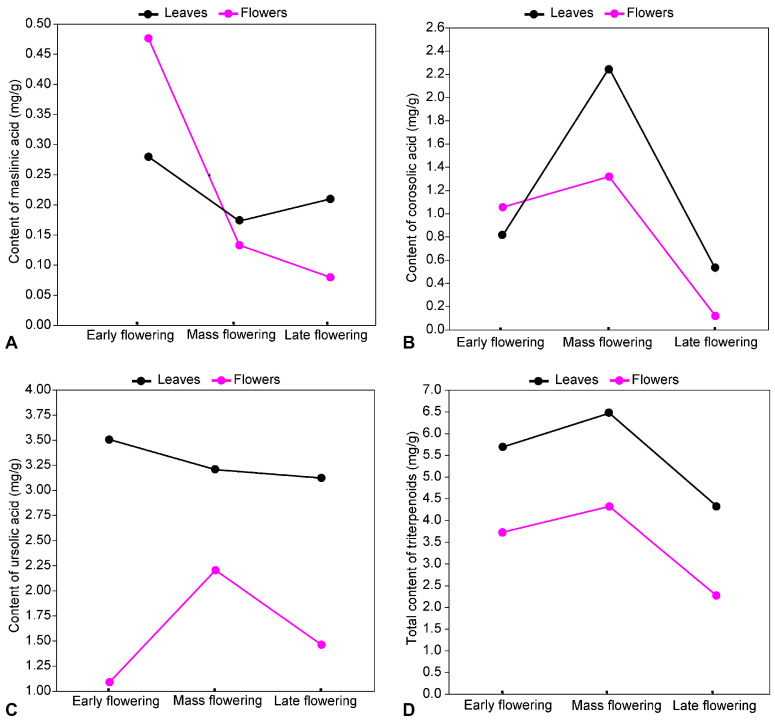
Dynamics of maslinic (**A**), corosolic (**B**), and ursolic (**C**) acids and the total content of triterpenoids (**D**) in *Epilobium angustifolium* flowers and leaves in different flowering phases.

**Figure 5 plants-13-00120-f005:**
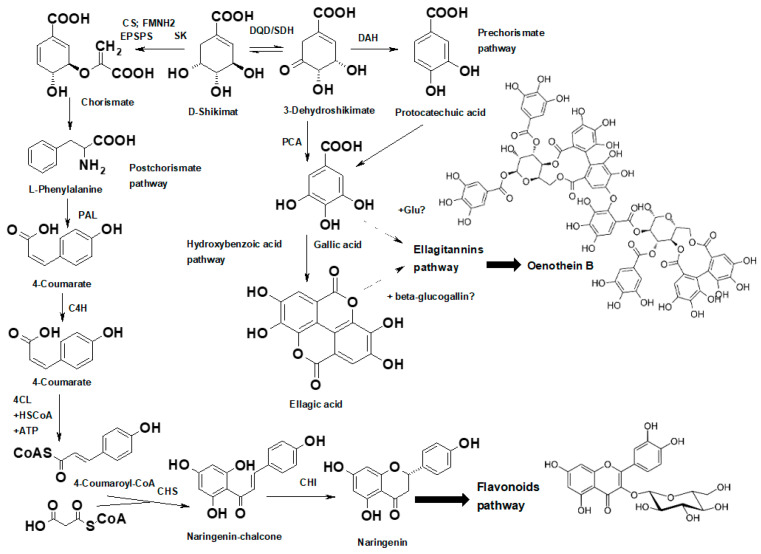
The proposed biosynthetic pathway of ellagitannins (oenothein B) in *E. angustifolium* (modified from data in [39,42]). The metabolic pathway starts with erythrose-4-phosphate and phosphoenolpyruvate (not shown in the figure). The enzymes that catalyse the individual steps from erythrose-4-phosphate and phosphoenolpyruvate are the following: 3-deoxy-D-arabino-heptulosonate-7-phosphate synthase, 3-dehydroquinate synthase, dehydroquinate dehydratase/shikimate dehydrogenase. Further, the biosynthetic steps involve: DQD/SDH—dehydroquinate dehydratase/shikimate dehydrogenase; DHS—dehydratase; SD—shikimate dehydrogenase; SK—shikimate kinase enzyme; EPSPS—5-enolpyruvylshikimate 3-phosphate synthase; CS—chorismate synthase; FMNH2—reduced flavin mononucleotide; PAL—phenylalanine ammonia lyase; C4H—cinnamate 4-hydroxylase; 4CL—4-coumaroyl-CoA ligase; CHS—chalcone synthase; CHI—chalcone isomerase; chorismate mutase prephenate aminotransferase; arogenate dehydrogenase; arogenate dehydratase; PCA—hydroxylase.

**Table 1 plants-13-00120-t001:** The total content of polyphenols and antioxidant effects of *Epilobium angustifolium* samples estimated with the UV method during early, mass, and late flowering (between 21 June and 14 July).

Indicator	Early Flowering		Mass Flowering			Late Flowering	
	Flowers	Leaves	Flowers	Leaves	Rhizomes	Flowers	Leaves
Total amount of polyphenols, mg GA/g dw	1.13 ± 0.02	1.29 ± 0.02	1.19 ±0.02	1.54 ± 0.03	0.56 ± 0.01	1.14 ± 0.02	1.17 ± 0.02
ABTS radical scavenging effects, μg TE/g dw	8744.44 ± 153.20	9755.56 ± 171.15	8750.44 ± 153.85	8655.56 ± 152.18	2577.78 ± 45.32	8742.41 ± 153.71	9865.10 ± 177.34
Ferric reducing antioxidant power, mg FE(II)/g dw	2287.50 ± 40.22	11,677.51 ± 205.32	3352.50 ± 58.95	9097.50 ± 159.96	2552.50 ± 44.87	3052.52 ± 53.67	3532.51 ± 62.11

**Table 2 plants-13-00120-t002:** Content of polyphenols and triterpenoids (mg/g dw) in *Epilobium angustifolium* samples in different flowering phases using the high-performance liquid chromatography photodiode array (HPLC–PDA) detection method (mean of three replications ± SD).

RT	Metabolite	Early Flowering		Mass Flowering			Late Flowering	
	Raw Material	Flowers	Leaves	Flowers	Leaves	Rhizomes	Flowers	Leaves
**Polyphenols**								
6.06	Gallic acid	0.60 ± 0.01	0.51 ± 0.01	0.77 ± 0.01	0.41 ± 0.01	0.14 ± 0.01	0.57 ± 0.02	0.64 ± 0.01
9.53	Neochlorogenic acid	0.73 ± 0.01	1.35 ± 0.02	1.54 ± 0.03	0.54 ± 0.01	0.01 ± 0.00	1.23 ± 0.03	1.99 ± 0.04
10.68	**Oenothein B**	39.70 ± 0.70	24.90 ± 0.44	50.26 ± 0.88	15.45 ± 0.27	1.95 ± 0.03	45.95 ± 0.81	54.62 ± 0.96
11.68	Chlorogenic acid	0.91 ± 0.02	1.61 ± 0.03	1.77 ± 0.03	1.10 ± 0.02	0.01 ± 0.00	1.55 ± 0.03	2.60 ± 0.05
13.04	**Oenothein A**	3.96 ± 0.07	4.40 ± 0.08	8.44 ± 0.15	1.85 ± 0.03	0.13 ± 0.01	7.26 ± 0.13	9.00 ± 0.16
22.72	Ellagic acid	0.41 ± 0.01	0.27 ± 0.01	0.51 ± 0.02	0.23 ± 0.01	0.17 ± 0.02	0.52 ± 0.02	0.41 ± 0.01
22.73	Rutin	0.18 ± 0.00	0.01 ± 0.00	1.19 ± 0.03	0.90 ± 0.02	-	1.56 ± 0.03	2.26 ± 0.04
23.65	Hyperoside	0.16 ± 0.00	0.29 ± 0.01	1.05 ± 0.02	0.80 ± 0.01	-	1.38 ± 0.02	2.00 ± 0.04
24.90	**Isoquercitrin**	4.77 ± 0.08	8.08 ± 0.14	7.98 ± 1.14	4.69 ± 0.08	0.05 ± 0.00	8.41 ± 0.15	11.18 ± 0.20
31.11	**Quercitrin**	2.75 ± 0.05	0.53 ± 0.01	0.72 ± 0.03	-	-	2.92 ± 0.05	0.56 ± 0.01
31.13	Quercetin	0.03 ± 0.00	0.04 ± 0.00	0.05 ± 0.01	0.02 ± 0.01	-	0.06 ± 0.00	0.08 ± 0.01
37.74	**Afzelin**	3.24 ± 0.06	0.41 ± 0.01	0.66 ± 0.02	-	-	6.47 ± 0.12	-
27.82	Guaijaverin	-	0.50 ± 0.02	1.63 ± 0.03	1.24 ± 0.02	-	0.99 ± 0.02	1.58 ± 0.03
Total phenolic acid content		0.66 ± 0.01	0.94 ± 0.03	1.15 ± 0.02	0.57 ± 0.01	0.08 ± 0.00	0.97 ± 0.03	1.41 ± 0.03
Total tannin content		21.83 ± 0.40	14.65 ± 0.26	29.35 ± 0.51	8.65 ± 0.15	1.04 ± 0.02	26.61 ± 0.47	31.81 ± 0.56
Total flavonoid content		1.86 ± 0.03	1.41 ± 0.02	1.90 ± 0.03	1.53 ± 0.03	0.05 ± 0.01	3.11 ± 0.06	2.94 ± 0.05
Total polyphenol content		57.44 ± 1.01	42.90 ± 0.75	75.85 ± 1.33	27.24 ± 0.48	2.45 ± 0.04	78.87 ± 1.39	86.92 ± 1.53
**Triterpenoids**								
6.278	Maslinic acid	0.50 ± 0.01	0.26 ± 0.01	0.13± 0.00	0.17 ± 0.01	0.10 ± 0.00	0.06 ± 0.00	0.21 ± 0.01
6.790	**Corosolic acid**	1.05 ± 0.02	0.81 ± 0.01	1.32 ± 0.02	2.23 ± 0.04	0.22 ± 0.01	0.12 ± 0.01	0.53 ± 0.01
12.06	Oleanolic acid	0.61 ± 0.01	0.54 ± 0.01	0.13 ± 0.00	0.50 ± 0.01	0.51 ± 0.01	0.19 ± 0.01	0.10 ± 0.00
12.69	**Ursolic acid**	1.08 ± 0.02	3.51 ± 0.06	2.20 ± 0.04	3.20 ± 0.06	0.11 ± 0.00	1.46 ± 0.03	3.12 ± 0.05
13.75	**Betulin**	0.51 ± 0.01	0.52 ± 0.02	0.53 ± 0.01	0.22 ± 0.01	0.08 ± 0.00	0.42 ± 0.01	0.38 ± 0.01
19.75	Erythrodiol	-	0.02 ± 0.00	-	0.02 ± 0.00	-	-	-
21.05	Uvaol	-	-	-	0.02 ± 0.00	-	-	-
	Total content	3.75 ± 0.07	5.66 ± 0.10	4.31 ± 0.08	6.36 ± 0.11	1.02 ± 0.02	2.25 ± 0.04	4.34 ± 0.08

‘-’ Not found.

**Table 3 plants-13-00120-t003:** The main validation characteristics of twelve phenolic reference compounds.

Metabolite	RT, min	Coefficient of Determination *R*^2 a^	Equation	Linearity Range (µg/mL)	LOD (µg/mL) ^b^	LOQ (µg/mL) ^c^	Repeatability RT/Area (%)	Precision RT/Area (%)
Gallic acid	6.10	0.99996	y = 3.39 × 10^4^x + 4.28 × 10^3^	0.679–174.000	0.154	0.662	0.2/0.7	0.3/0.8
Neochlorogenic acid	9.53	0.99982	y = 5.50 × 10^4^x − 2.29 × 10^3^	0.195–25.000	0.102	0.307	0.3/0.7	0.4/0.8
Oenothein B	10.66	0.99764	y = 1.81 × 10^4^x − 9.31 × 10^3^	0.781–100.000	0.210	0.424	0.1/0.8	0.3/1.2
Chlorogenic acid	11.95	0.99980	y = 3.10 × 10^4^x − 7.62 × 10^3^	0.406–208.200	0.049	0.147	0.2/0.6	0.4/0.8
Ellagic acid	22.48	0.99999	y = 1.00 × 105x − 6.18 × 10^3^	0.185–23.700	0.067	0.126	0.1/0.3	0.2/0.5
Rutin	22.80	0.99993	y = 1.85 × 10^4^x − 5.68 × 10^3^	2.456–78.600	0.200	0.731	0.3/0.7	0.4/0.8
Hyperoside	23.62	0.99985	y = 2.10 × 10^4^x − 5.17 × 10^3^	1.514–193.800	0.126	0.378	0.2/0.6	0.3/0.8
Isoquercitrin	24.50	0.99978	y = 2.18 × 10^4^x − 7.31 × 10^3^	1.462–187.200	0.278	0.772	0.2/0.7	0.4/0.9
Guaijaverin	28.31	0.99838	y = 1.21 × 10^4^x + 5.33 × 10^3^	1.562–100.000	0.260	0.781	0.1/1.0	0.3/1.2
Quercitrin	30.90	0.99991	y = 1.98 × 10^4^x − 9.22 × 10^3^	0.802–205.500	0.134	0.401	0.3/0.9	0.4/1.0
Afzelin	37.95	0.99981	y = 9.52 × 10^3^x + 7.46 × 10^3^	0.781–100.000	0.155	0.390	0.8/0.9	0.9/1.0
Quercetin	43.65	0.99994	y = 3.61 × 10^4^x − 1.42 × 10^3^	1.025–131.200	0.162	0.995	0.1/0.5	0.2/0.8

^a^ Concentration of metabolites (mg/mL); y, peak area; ^b^ LOD, limit of detection (S/N = 3); ^c^ LOQ, limit of quantification (S/N = 10).

## Data Availability

All data are included in the article.

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
