# Peer review of "Phenological Variations in the Content of Polyphenols and Triterpenoids in Epilobium angustifolium Herb Originating from Ukraine"

_plants, 2023, doi:10.3390/plants13010120_

Round 1
Reviewer 1 Report
Comments and Suggestions for Authors
Abstract. The main triterpenoid saponins have been mentioned, but not the main polyphenols. Which are the principal polyphenols in the herbal drug? How many polyphenols and triterpenoids have been identified by HPLC?
Introduction. Line 49: include citation.
Line 51: Check the presentation of the citation style.
Line 54: Must be "available" instead of "avaiabe".
Lines 58-59: The sentence "Plants of the genus Epilobium L. are a rich source of secondary metabolites, especially 58 polyphenols, including flavonoids, phenolic acids, amino acids and tannins..." needs improvement because amino acids do not belong to the group of polyphenols.
What about clinical trials of E. angustifolium?
Results.
Lines 97-98: The first sentence belongs to the Methodology section, this is not the result.
Subsections 2.3.1-2.3.4: Which tables/figures show the results? No citations.
Line 252: Must be "2.6.2. Quantitative Dynamics of Triterpenoids" instead of "2.6.2. Quantitative Dynamics of Triterpenoid"
Lines 260-263: No need to give four decimal places after the comma (0.1337), three places are enough (0.134).
Discussion. The Figure 3 is presented after the Figure 5. Correct the order of figures.
Material and Methods
Line 485: Use italics in the plant's Latin name.
Line 469: No need to repeat the Latin synonym of the plant.
Lines 469-476: What is the number of the voucher specimen and where is it available?
Line 507: No need to write "GmbH",
References. The list of References does not correspond to the style of the journal Plants. It needs improvement according to instructions for the authors.
General comment: check technical errors in the text.
Author Response
We are grateful to the reviewer for his/her work and any comments. We have made all the corrections and additions, removed repetitions in the text, and also checked the language well. Answers to each comment are in the file below. Thank you.

Reviewer 2 Report
Comments and Suggestions for Authors
Introduction
Lines 45, 48, 49, 52: lack reference.
Line 54: check the spelling of ‘avaiabe’.
Lines 53-56: delete or break the sentence and rewrite clearly with reference.
Line 58: Delete ‘Plants of the’ and ‘secondary metabolites, especially’.
Lines 62-61: lack reference.
Lines 69-70: delete.
Rewrite third and fourth paragraph (literature review), the writing is too haphazard and repetitive pharmacological activities, write in simpler way so that it will be easily understandable and do the literature review related to your work only.
I am also unsatisfied with your aim of the study (the last paragraph). Your purpose of the study should be clear.
Results
Write full-form of the technical terms while writing for the first time for e.g. μg/g DW, GA/g, HPLC, HPLC-DAD etc.
Discussion
Lines 292-311: not necessary, delete.
Rewrite the discussion section because please do not write about other pharmacological properties because your research is on antioxidant properties only why are you mentioning pharmacological properties repetitively. Discussion section is unnecessarily too much elaborated.
Comments on the Quality of English LanguageAfter addressing the review comments, English language correction must be done.
Author Response

(The authors gave the same response as above.)

Round 2
Reviewer 1 Report
Comments and Suggestions for Authors
Dear Editor,
The manuscript has been modified according to my comments. We can accept it as it is.
Reviewer 2 Report
Comments and Suggestions for Authors
hello
You have addressed my review comments. English language correction should be done before publication.
best regards
Comments on the Quality of English Languagehello
English language correction should be done before publication.
best regards